# The Deletion of *LeuRS* Revealed Its Important Roles in Osmotic Stress Tolerance, Amino Acid and Sugar Metabolism, and the Reproduction Process of *Aspergillus montevidensis*

**DOI:** 10.3390/jof10010036

**Published:** 2024-01-03

**Authors:** Xiaowei Ding, Wanting Liu, Kaihui Liu, Xiang Gao, Yue Liu

**Affiliations:** School of Food and Biological Engineering, Shaanxi University of Science and Technology, Xi’an 710021, Chinaliuyue02023@163.com (Y.L.)

**Keywords:** leucyl-tRNA synthetase, transcriptome, metabolome, *Aspergillus montevidensis*, osmotic tolerance

## Abstract

*Aspergillus montevidensis* is an important domesticated fungus that has been applied to produce many traditional fermented foods under high osmotic conditions. However, the detailed mechanisms of tolerance to osmotic stress remain largely unknown. Here, we construct a target-deleted strain (Δ*LeuRS*) of *A*. *montevidensis* and found that the Δ*LeuRS* mutants grew slowly and suppressed the development of the cleistothecium compared to the wide-type strains (WT) under salt-stressed and non-stressed conditions. Furthermore, differentially expressed genes (*p* < 0.001) governed by *LeuRS* were involved in salt tolerance, ABC transporter, amino acid metabolism, sugar metabolism, and the reproduction process. The Δ*LeuRS* strains compared to WT strains under short- and long-term salinity stress especially altered accumulation levels of metabolites, such as amino acids and derivatives, carbohydrates, organic acids, and fatty acids. This study provides new insights into the underlying mechanisms of salinity tolerance and lays a foundation for flavor improvement of foods fermented with *A*. *montevidensis*.

## 1. Introduction

The filamentous fungi *Aspergillus* is a complex group of ascomycetes comprising about 340 officially recognized species, many of which are widely used in food fermentation [1]. Among *Aspergillus* species, *A*. *montevidensis* is an important domesticated fungus that has been used to produce or isolate from fermented and ripened katsuobushi [2]. *A*. *amstelodami* (the anamorph of *A*. *montevidensis*) is commonly isolated from meju, a brick of fermented dried soybeans, and frequently occurs in fermented cocoa beans [3,4]. Furthermore, these species dominate the microbial community involved in post-fermented Chinese dark teas, including Pu-erh tea, Fu brick tea, and Liupao tea [5,6,7]. During the production processes of these foods, high osmolarity conditions enhance the dominance of *A. montevidensis*, which can result in improved quality and taste of fermented foods. Therefore, it is of great significance to understand the osmophilic mechanism of *A*. *montevidensis*.

We previously found that *A*. *montevidensis* is highly dependent on the fine regulation of morphological, transcriptional, and metabolic responses to manage osmotic stress tolerance [8]. Under salinity stress, *A. montevidensis* widely developed yellow fruiting bodies (cleistothecia), which are known as ‘golden flowers’ in Chinese brick tea, and hypha growth was obviously promoted [9]. Salinity-induced mycelia significantly expressed hundreds of genes that controlled cellular processes, such as oxidative stress response, amino acid transport and metabolism, glycolysis and TCA cycles, and fatty acid β-oxidation, for the intracellular accumulation of a variety of amino acids, soluble sugars, and fatty acids. Among these significantly expressed genes, a unigene encoding leucyl-tRNA synthetase (*LeuRS*), which routinely catalyzes the specific attachment of leucine to tRNA^Leu^ in protein synthesis [10], robustly increased its expression by a factor of 11.52 [8], suggesting that *LeuRS* could influence osmotic stress tolerance and the reproduction process of *A. montevidensis*.

Sexual and asexual development in *Aspergillu*s species is a highly complex biological process which is affected by both environmental factors and complex intrinsic signals [11]. Extensive studies in *A*. *nidulans* showed that the sexual cycle was influenced by many environmental variables, such as light, pH values, temperature, and atmospheric gases [12]. To date, at least 78 genes are required for the modulation of the sexual process of *Aspergillus*. However, there are 20 regulators controlling hundreds of genes, which are involved in asexual development of *A*. *nidulans* [13]. These sex-related genes are linked to perception of environmental signals, signal transduction, transcriptional activators, and maturation of ascospores and conidiophores [11].

Recent studies have shown many previously unknown functions of *LeuRS*; for example, it serves as a leucine sensor for the mTORC1 signaling pathway in eukaryotes, which subsequently regulates protein synthesis, cell growth, ribosome biogenesis, nutrient uptake, and autophagy [14,15,16]. *LeuRS* also mediates tolerance to glucose starvation and norvaline-induced stress in yeast and mammals [17,18]. Here, we found that deletion of *LeuRS* in *A*. *montevidensis* influenced mycelial growth rate, antioxidant capacity, significantly suppressed the formation of cleistothecia, and prompted conidial development under salt stress. Δ*LeuRS* mutants changed expression patterns of multiple genes for substance transport, as well as amino acid and sugar metabolism, etc. Together, our findings highlight that *LeuRS* plays a key role in the salinity tolerance of *A. montevidensis*.

## 2. Materials and Methods

### 2.1. Fungal Strains and Culture Conditions

The wide-type *A*. *montevidensis* (no: CGMCC 3.15762) used in this study was maintained routinely on yeast peptone dextrose agar slants (YPD: 1% yeast extract, 2% tryptone, 2% glucose, 1.5% agar, pH 6.5), supplemented with the final NaCl concentration (1.5 M). The morphological and growth rates of the WT and Δ*LeuRS* mutants were examined on YPD media with 0 M and 1.5 M NaCl at 28 °C. The WT and mutants were statically cultured in YPD liquid medium with 1.5 M NaCl at 28 °C for 14 days (long-term salt stress, WT_LS and Δ*LeuRS*_LS). These strains were also grown in YPD liquid medium without salt at 28 °C for 14 days and then stressed by the addition of salt (1.5 M) for 1 h (short-term salt stress, WT_SS and Δ*LeuRS*_SS). Control samples (WT_C and Δ*LeuRS*_C) were obtained from 14-day cultures untreated with salt. Salt-treated and untreated cultures were washed twice with a pre-chilled PBS solution. Mycelia were collected using centrifugation at 4 °C, 12,000 rpm for 10 min, and used in the examination of transcriptome, metabolome, and antioxidant enzyme activity (catalase, CAT; superoxide dismutase, SOD) [19].

### 2.2. Constructs for LeuRS Knockdown and Fungal Transformation

We sequenced the whole genome of *A. montevidensis* and found a 3501-bp single-copy gene, which encodes *LeuRS* [8,20]. The LeuRS gene of *A*. *montevidensis* (WT) was eliminated by homologous recombination. Briefly, the flanking fragments upstream and downstream (around 1500 bp) of the LeuRS gene were amplified from the wide-type strain, with the specific primer pairs *LeuRS1/1F-1R* (*LeuRS1/1F* (PstI): 5′-GCCTGCAGTCCCGATCTTTCACAGACTG-3′; *LeuRS1/1R* (XbaI): 5′-TCTCTAGACCGAAGAACTTGGGGTACTT-3′) and *LeuRS2*/*2F*-*2R* (*LeuRS2*/*2F* (SpeI): 5′-ACACTAGTGATGCTACTCGTATCGCTTT-3′; *LeuRS2*/*2R* (EcoRV): 5′-AGGATATCGGAACAGACATAGCGGTTTG-3′), respectively. The amplified DNA fragments were digested by PstI and XbaI, and SpeI and EcoRV, respectively, and then ligated into the plasmid pPK2 at the corresponding restriction sites. The construct pPK2-LeuRS1-Sur-LeuRS2 was transformed into the wide-type strain according to *Agrobacterium tumefaciens*-mediated transformation [21]. Transformants were selected on YPD agar containing 100 μg/mL of chlorimuron. The colonies were confirmed using PCR amplification using the primer pairs (*LeuRS*-YZ5f: 5′-CCGTTTTCGCTTTGGATGAG-3′; *LeuRS*-YZ3r: 5′-TGACTCCTGACTCATACTGC-3′) and sequenced.

### 2.3. Transcriptome Sequencing and Data Analysis 

RNA extraction and library preparation of salt-treated and untreated cultures were performed as described by Ding et al. [8]. The cDNA libraries were sequenced on an Illumina Novaseq 6000 platform and 150-bp paired-end reads were generated. The raw data were processed by removing adaptor sequences, low quality reads, and reads with unknown nucleotides > 5%. High-quality read assembly was performed using Trinity software (v2.4.0) [22]. With three biological replicates, transcriptomic sequencing generated 383,214,494 and 399,020,304 clean reads for the three comparison groups (18 libraries). Clean reads were assigned to the reference genome (accesssion number JAJFZZ000000000) [20]. The RPKM (reads per kb per million reads) of each unigene was calculated to analyze the expression level of genes between samples. Genes with a false discovery rate (FDR) < 0.001 and |log2 ratio| ≥ 1 were identified as differentially expressed genes (DEGs) [23]. Principal component analysis (PCA) was performed using the R package (version 4.0.0). DEG enrichment analysis was performed in different databases using BLASTx, including NCBI nonredundant protein (NR), Gene Ontology (GO), Swissprot, KOG, and the Kyoto Encyclopedia of Genes and Genomes (KEGG) with E < 10^−5^. The FPKM values of DEGs were used in a heatmap analysis. The biochemical pathways of DEGs were further implemented by KEGG mapping using the cluster Profiler (v3.4.4) [24].

### 2.4. UHPLC-Q-TOF/MS of Metabolites and Data Analysis

The salinity-treated and non-treated freeze-dried fungal mycelia (0.1 g) of the WT and the mutant strains (Δ*LeuRS*) were extracted with methanol-acetonitrile-water (2:2:1, *v*/*v*), which contained internal standard mixtures labeled with isotopes. The samples were vortexed, sonicated in an ice water bath for 4 min, incubated at −40 °C for 1 h, and then centrifuged at 4 °C for 15 min (12,000 rpm) to collect the supernatants. The extraction process was repeated three times. The supernatants were pooled and filtered through a 0.22-µm filter. Meanwhile, a quality control (QC) sample was prepared by mixing equal volumes of each extraction, and was injected to monitor the stability of the system. Metabolites were analyzed using an UHPLC system (Vanquish, Thermo Fisher Scientific, Pleasanton, CA, USA) with a UPLC BEH Amide column (2.1 mm × 100 mm, 1.7 μm) coupled with a Q Exactive HFX mass spectrometer (Orbitrap MS, Thermo Fisher). The mobile phase consisted of water (pH 9.75) (containing 25 mmol/L ammonium acetate and 25 mmol/L ammonia hydroxide) and acetonitrile. The autosampler temperature was 4 °C and the injection volume was 3 μL. The elution gradient was carried out as follows: 0 min, 5% B; 3 min, 20% B; 9 min, 95% B; 13 min, 95% B; 13.1 min, 5% B; and 16 min, 5% B. A mass scan range of *m*/*z* 70–1000 was performed for full-scan analysis. The ESI source conditions were established as follows: sheath gas flow rate 30 arb, aux gas flow rate 25 arb, capillary temperature 350 °C, full MS resolution 60,000, collision energy 10/30/60, and spray voltage 3.6 (positive) or −3.2 kV (negative), respectively. The raw mass data were converted into mzXML format using ProteoWizard software (v3.0.9134). XCMS software (v3.7.1) was applied for peak extraction, peak alignment, and peak matching. Metabolites were then annotated using BiotreeDB (v2.1) software.

### 2.5. Statistical Analysis

All experiments were conducted in independent triplicates, and data are reported as the mean value ± SD (standard deviation). Student’s *t*-test was performed to examine the significant differences in metabolites between groups (*p* < 0.05) by using SPSS 26.0 software (SPSS Inc., Chicago, IL, USA). Differential metabolites with absolute log2FC ≥ 1 and *p* < 0.05 were selected for further analysis. Orthogonal projections to latent structures discriminant analysis (OPLS-DA) using SIMCA 14.1 [25] were conducted to screen characteristic components contributing to group discrimination. Differential metabolites were selected by the statistically significant variable importance threshold in projection values ≥1 and *p*-values < 0.05. Significantly different data sets in transcriptomic and metabolomics analysis were used to generate metabolite-transcript pathways by KEGG mapping.

### 2.6. Accession Number(s)

The raw sequence data have been submitted to the National Center for Biotechnology Information (NCBI) Sequence Read Archive (SRA) under BioSample accessions: SRR24694851 to SRR24694851.

## 3. Results 

### 3.1. LeuRS Influences Mycelial Growth, Cleistothecium Formation, and Stress Tolerance of A. montevidensis

Δ*LeuRS* mutants were verified using PCR amplification and sequencing (Appendix A). Mycelial growth showed a morphological difference between the WT strain and the *LeuRS* mutant on the YPD medium supplemented with or without NaCl. The WT strain had yellow colonies, while the Δ*LeuRS* colonies were white under salinity or non-salt stress for 14 days (Figure 1). The WT strain also produced a large number of yellow-colored spherical cleistothecia (Figure 1b,c), while the formation of cleistothecium was strongly inhibited in the *LeuRS* mutant (Figure 1e,f). The deletion of *LeuRS* resulted in a significant reduction (*p* < 0.001) of the growth rates of the hyphal compared to WT strains under salinity or non-salt conditions (Figure 1k). Salt stress causes oxidative stress in living cells [26]. Therefore, we examined the antioxidative activities of the *LeuRS* mutant and the WT strain. CAT activity in the Δ*LeuRS* cultures induced with or without NaCl was significantly (*p* < 0.05) lower than in WT, and SOD activity in mutant samples suffering from long-term salinity stress was obviously lower compared to the corresponding WT group (Figure 1l). These results indicated that *LeuRS* plays a crucial role in the regulation of mycelial growth, antioxidative enzyme activities, and sexual development of *A. montevidensis*.

### 3.2. Transcriptome Profiles of the LeuRS Mutant and the WT Strain

We matched unigene sequences with the NR, Swiss-prot, KEGG, and KOG databases using blastx (E < 10^−5^). Of these, a total of 7165 unigenes were functionally annotated. PCA indicated that all biological replicates were grouped together, and PC1 and PC2 captured 60% of the variance in the data. This represents the difference in gene expression profiles between the WT strain and the Δ*LeuRS* mutant (Figure 2a). We identified significantly differentially expressed unigenes (DEGs) using FDR < 0.001 and the |log_2_ ratio| ≥ 1 as the threshold. Among the genes significantly expressed up, there were 82 DEGs in the Δ*LeuRS*_SS/Δ*LeuRS*_C and 1107 in the Δ*LeuRS*_LS/Δ*LeuRS*_C, with 50 DEGs shared by these two groups (Figure 2b). There were 153 DEGs in the WT_SS/WT_C and 881 in the WT_LS/WT_C, with 62 co-occurred DEGs (Figure 2c). Among the significantly down-expressed genes, there were 79 DEGs in the Δ*LeuRS*_SS/Δ*LeuRS*_C and 796 in the Δ*LeuRS*_LS/Δ*LeuRS*_C, with 51 common DEGs (Figure 2d). There were 576 DEGs in the WT_SS/WT_C and 1070 in the WT_LS/WT_C, with 370 common DEGs (Figure 2e). Furthermore, more DEGs responded to long-term saline conditions than to short-term treatment, but the majority of genes in response to short-term treatment also responded to long-term salt stress (Figure 2b,d,e). Our results showed that *LeuRS* was of significance in mediating gene expression in *A*. *montevidensis* in response to change in salinity.

### 3.3. Analysis of DEGs

KEGG pathway analysis of DEGs revealed that more metabolic processes in Δ*LeuRS* and WT were influenced by long-term salt stress than short-term saline shock (Figure 3). For example, many DEGs up-regulated (*p* < 0.05) in Δ*LeuRS*_LS/Δ*LeuRS*_C participated in 13 pathways (fructose and mannose metabolism, tyrosine metabolism, alanine, aspartate and glutamate metabolism, arginine and proline metabolism, ABC transporters, etc.). However, a few DEGs were associated with three pathways in Δ*LeuRS*_SS/Δ*LeuRS*_C. Furthermore, the number of DEGs expressed up, involved in alanine, aspartate, and glutamate metabolism and nitrogen metabolism, increased in Δ*LeuRS* as stress continued. The DEGs descending in expression were mainly related (*p* < 0.05) to four KEGG pathways (biosynthesis of amino acids, peroxisome, butanoate metabolism, etc.) in Δ*LeuRS*_LS/Δ*LeuRS*_C. Similarly, many DEGs up-regulated in WT_LS/WT_C were strongly enriched in (*p* < 0.05) ribosome biogenesis, pentose and glucuronate interconversions, and sphingolipid metabolism, while a few genes participated in oxidative phosphorylation in WT_SS/WT_C. And the decreased DEGs in WT_LS/WT_C were enriched (*p* < 0.05) in 11 pathways, which were indispensable to starch and sucrose metabolism, glycine, serine and threonine metabolism, and arginine and proline metabolism, etc. (Figure 3d). 

### 3.4. DEGs Significantly Related to ABC Transporters, Nitrogen and Carbon Metabolism, and Reproduction

To understand the transcriptional changes of DEGs of Δ*LeuRS* and WT in response to salt stress, response-specific DEGs were selected for heatmap analysis (Figure 4, Appendix A). We found that most DEGs in Δ*LeuRS*_C, Δ*LeuRS*_SS, and Δ*LeuRS*_LS exhibited distinct expression patterns, compared to their transcriptional levels in the corresponding WT samples. For example, the expression two unigenes encoding ABC transporters was dramatically decreased in Δ*LeuRS*_C and Δ*LeuRS*_SS and increased in Δ*LeuRS*_LS, while the reverse trend was shown in WT_C, WT_SS, and WT_LS. Genes homologous to adenylosuccinate synthase, aspartate aminotransferase, and 1-pyrroline-5-carboxylate dehydrogenase (which were involved in alanine, aspartate, and glutamate metabolism) exhibited lower expression levels in mutants than in WT strains with different osmotic conditions. Gene-coding enzymes such as ornithine carbamoyltransferase, glutamate dehydrogenase, arginase, and argininosuccinate lyase, which control arginine biosynthesis, were strongly down-regulated in Δ*LeuRS* groups, while these had high transcriptional levels in WT. Similar expression patterns of genes were also observed in tryptophan metabolism. Genes related to sugar metabolism and glyoxylate and dicarboxylate metabolism, such as malate dehydrogenase, 6-phosphogluconate dehydrogenase, and glycine hydroxymethyltransferase were increased in expression in the three treatment groups of Δ*LeuRS* but strongly inhibited in WT. We found that some regulatory factors, which regulate amino acid metabolism, sexual reproduction, and tolerance to osmotic stress, were down-regulated in mutants. For example, several genes encoding Zn (II) 2 Cys6 (C6) transcription factors exhibited lower expression levels in Δ*LeuRS* against them in WT. Genes (such as *MAT1*, *Ste20*, and *VeA*) involved in sexual reproduction were obviously suppressed in Δ*LeuRS* compared to those in WT. Similarly, mitogen-activated protein kinase (HOG1) was also negatively regulated in Δ*LeuRS*. We observed that Δ*LeuRS*_LS augmented the expression of many genes more than Δ*LeuRS*_C and Δ*LeuRS*_SS. In addition, we also compared the expression changes of DEGs in Δ*LeuRS* with them in WT under the same osmotic stress conditions. We found that most DEGs of Δ*LeuRS* were down-regulated in expression and that the expression levels of many genes increased as the osmotic conditions continued. Our findings demonstrated that LeuRS mediated many cellular processes (substance transport, amino acid and sugar metabolism, and reproduction) of the fungus under salt stress.

### 3.5. Analysis of Differential Metabolites

Untargeted UHPLC-Q-TOF/MS analysis identified a total of 366 compounds, and PCA revealed significant differences in metabolites responsible for separation between the Δ*LeuRS* and WT groups (Appendix A), indicating that these two samples had substantially different metabolite profiles. We identified 72 discriminatory metabolites of Δ*LeuRS*_SS/Δ*LeuRS*_C, 77 of Δ*LeuRS*_LS/Δ*LeuRS*_C, 92 of WT_SS/WT_C, and 91 of WT_LS/WT_C, based on a VIP threshold of ≥1 (Appendix A). We further detected 23 and 43 compounds which were only identified in Δ*LeuRS*_SS/Δ*LeuRS*_C and WT_SS/WT_C, respectively. Unique metabolites in Δ*LeuRS*_SS/Δ*LeuRS*_C, such as N-acetyl-L-phenylalanine (log2FC = −1.6), 2-hydroxybutyric acid (log2FC = −2.2), and gentisic acid (log2FC = −2.9) were down-regulated (Figure 5a). Among the special substances in WT_SS/WT_C, the relative abundances of sucrose (log2FC = 2.2), 4-hydroxycinnamic acid (log2FC = 3.3), and kynurenic acid (log2FC = 12.0) increased, which are related to starch and sucrose metabolism, tyrosine metabolism, and tryptophan metabolism, respectively (Appendix A). However, substances, such as 4-hydroxyproline (log2FC = −1.1), L-methionine (log2FC = −1.6), L-phenylalanine (log2FC = −1.5), and L-tryptophan (log2FC = −1.6) decreased their levels in WT_SS/WT_C, which participate in arginine and proline metabolism, cysteine and methionine metabolism, phenylalanine, tyrosine and tryptophan biosynthesis, and tryptophan metabolism (Appendix A). The majority of common molecules in Δ*LeuRS*_SS/Δ*LeuRS*_C and WT_SS/WT_C decreased (Figure 5b). Furthermore, we observed that D-Mannose 1-phosphate (log2FC = −3.7), gluconic acid (log2FC = −2.6), and gluconolactone (log2FC = −3.7) decreased more in Δ*LeuRS*_SS/Δ*LeuRS*_C than in WT_SS/WT_C, suggesting that sugar metabolism was enhanced in Δ*LeuRS*_SS. In particular, WT_SS/WT_C strongly down-regulated 4-hydroxyphenylpyruvate (log2FC = −14.3) but increased 4-hydroxycinnamic acid (log2FC = 3.3), indicating that carbon flow from 4-hydroxyphenylpyruvate to 4-hydroxycinnamic acid in tyrosine metabolism could improve the salt tolerance of WT strains. 

Moreover, we identified 32 and 46 compounds that occurred specially in Δ*LeuRS*_LS/Δ*LeuRS*_C and WT_LS/WT_C, respectively. Pyruvic acid (log2FC = 1.0), 4-hydroxycinnamic acid (log2FC = 2.2), and 3-hydroxyphenylacetic acid (log2FC = 2.9) in Δ*LeuRS*_LS/Δ*LeuRS*_C were respectively related to alanine, aspartate and glutamate metabolism, and tyrosine metabolism (Figure 5c, Appendix A). The characteristic compounds in WT_LS/WT_C, such as dulcitol, sucrose, and D-mannose 1-phosphate increased their relative contents from log2FC = 1.0 to log2FC = 1.9, indicating that sugar metabolism was slowed for salt adaptation. 2-Ketobutyric acid (log2FC = 1.1), N-(2-furoyl) glycine (log2FC = 2.6), and 3-methyladipic acid (log2FC = 3.9) were up-regulated in this group, which were associated with glycine, serine and threonine metabolism, and alanine, aspartate, and glutamate metabolism (Appendix A). However, DEGs of these pathways in WT_LS/WT_C were down-regulated (Figure 3d). These suggested that glycine, serine and threonine metabolism, and alanine, aspartate, and glutamate metabolism could be decelerated in the fungus in response to long-term salinity. In comparison with Δ*LeuRS*_LS/Δ*LeuRS*_C, m-coumaric acid (log2FC = −4.9) decreased in WT_LS/WT_C. Among the co-occurring compounds, the sugars were more negatively regulated in Δ*LeuRS*_LS/Δ*LeuRS*_C than in WT_LS/WT_C (Figure 5d). However, WT_LS increased its relative abundances of kynurenic acid (log2FC = 11.8) in tryptophan metabolism and decreased the levels of 4-hydroxyphenylpyruvate (log2FC = −14.2) in tyrosine metabolism. 

## 4. Discussion 

*A*. *montevidensis* is commonly used in the production of traditional fermented foods [4,7]. During fermentation processes, the transcriptional responses of *A*. *montevidensis* to osmotic conditions are necessary for the flavor formation of foods. To reveal the functions of *LeuRS* (which were strikingly expressed higher in *A*. *montevidensis* under osmotic stress), we constructed a knockout mutant of *LeuRS* and found that the deletion of *LeuRS* inhibited growth both in the presence and absence of NaCl and deactivated the formation of cleistothecium compared to the WT strain (Figure 1). We also found another *LeuRS* (which was designated as *mLeuRS*, encoding a mitochondrial leucyl-tRNA synthetase) in the fungus. The average FPKM values of *mLeuRS* were higher in Δ*LeuRS*_C (25.7 FPKM) and Δ*LeuRS*_SS (24.9 FPKM) than in WT_C (5.0 FPKM) and WT_SS (6.2 FPKM), but lower in Δ*LeuRS*_LS (5.2 FPKM) compared to WT_LS (11.1 FPKM). This could be the reason why the knockout of *LeuRS* in the fungus did not cause lethality. Transcriptomic and metabolomic analyses revealed that *LeuRS* are closely related to ABC transporters, amino acid metabolism, carbohydrate metabolism, and sexual development of *A. montevidensis*.

Here, we observed that DEGs associated with ATP-binding cassette (ABC) transporters were down-regulated in Δ*LeuRS*, while they were obviously increased in WT samples (Figure 4, Appendix A). Long-term saline induction increased the transcriptional levels of ABC transporters in mutants. Extensive research showed that ABC transporters are evolutionarily conserved integral membrane proteins responsible for the allocation of a wide variety of substrates, including ions, sugars, amino acids, polypeptides, complex lipids, toxic metabolites, and even toxins [27,28]. Previous reports also indicated that the specific upregulation of some ABC transporters was involved in cell osmo-stress tolerance [29,30].

Specific regulation of amino acid metabolism is crucial to controlling cell growth and proliferation, and osmotic adjustment of all living organisms [8,31,32]. As expected, 34 significant DEGs between Δ*LeuRS* and WT groups under short- and long-term salt stress were associated with the metabolism pathways of many amino acids (Appendix A). We found that many up-expressed genes of Δ*LeuRS* mutants with short- and long-term saline stresses, compared to controls (Δ*LeuRS_*C*)*, were significantly enriched in four KEGG pathways of amino acid metabolism (alanine, aspartate, and glutamate metabolism, arginine and proline metabolism, and tryptophan metabolism, etc.). In contrast, the decreased expression of DEGs between the WT_LS vs. WT_C groups were mainly enriched in alanine, aspartate, and glutamate metabolism, and arginine and proline metabolism. Furthermore, we observed that Δ*LeuRS_*LS up-expressed DEGs in mitophagy (Figure 3a), suggesting that the recycling of nutrients to avoid carbon and nitrogen starvation may be an important survival strategy for the fungus with long-term cultivation under osmotic stress. 

We found that Δ*LeuRS* mutants exposed to salt stress increased the expression of genes encoding glutamine synthetase and aspartate carbamoyl transferase, which were associated with alanine, aspartate, and glutamate metabolism. These proteins were up-regulated in salt-induced plants [33,34]. However, Δ*LeuRS* with or without salt significantly inhibited the transcription of gene-encoding enzymes such as aspartate aminotransferase, succinate-semialdehyde dehydrogenase, and 1-pyrroline-5-carboxylate dehydrogenase. Previous reports showed that up-expression of aspartate aminotransferase and 1-pyrroline-5-carboxylate dehydrogenase was important for the survival of plants under saline conditions [33,35]. Furthermore, we observed that WT strains under long-term saline conditions increased the levels of some amino acids and organic acids (e.g., DL-phenylalanine, N-(2-furoyl) glycine, and 3-methyladipic acid) (Figure 5), which could contribute to the salt adaptation of the fungus. 

Previous studies indicated that the activation of arginine and proline metabolism confers enhanced stress tolerance in different eukaryotes [8,36]. However, little is known about the regulatory mechanism of the pathway. Here, we found that DEGs were significantly enriched in arginine and proline metabolism (Figure 4, Appendix A). The expression of genes encoding ornithine decarboxylase and the S-adenosylmethionine decarboxylase proenzyme increased in Δ*LeuRS_*LS and WT_LS, compared to their controls. These genes participated in the biosynthesis of polyamines, which can improve plant tolerance to salinity by scavenging free radicals [37]. A gene-encoding D-amino acid oxidase was up-expressed in Δ*LeuRS_*LS/Δ*LeuRS_*C, which accelerates the process of proline transformation into 1-pyrroline-2-carboxylate in proline metabolism. A previous study showed that the increase in D-amino-acid oxidase activity was correlated with the heat tolerance of fungi [38]. Interestingly, the expression of all genes (such as carbamoyl-phosphate synthase, arginase, and argininosuccinate synthase) in the ornithine cycle was strongly down-regulated in Δ*LeuRS* groups, while they maintained high transcriptional levels in WT samples. In addition, two gene-encoding arginase and argininosuccinate synthase in mutants were fully blocked. The deletion of carbamoyl-phosphate synthase in *Colletotrichum gloeosporioides* led to a slow growth rate and an extreme sensitivity to high osmotic stress [39]. Up-regulation of arginase could alleviate damage to plants caused by salinity [40]. These suggested that *LeuRS* could mediate the ornithine cycle in the fungal adaptation to salinity. Moreover, we observed that some genes in tryptophan metabolism also maintained relatively low expression in *LeuRS*_C and *LeuRS*_SS, but they were enhanced in *LeuRS*_LS. A similar case in the plant showed that activation of the phenylalanine, tyrosine, and tryptophan pathways improved osmoprotective ability [41]. Δ*LeuRS* under salt stress specially modulated the metabolic flow of some amino acids, which might rescue mutant defects in terms of salt tolerance. 

Sugar metabolism has been shown to play a key role in maintaining osmotic homeostasis and intracellular energy balance through carbon partitioning in microbes [42]. Here, we observed that many DEGs in Δ*LeuRS* maintained relatively low expression, which encoded enzymes involved in the glycolysis/glucoseogenesis, citrate cycle, and glyoxylate and dicarboxylate metabolism (Appendix A). Hexokinases control the first step of all major pathways of glucose utilization and therefore influence the extent and direction of glucose flux within the cell. The expression of hexokinases promoted tolerance to drought and salt stress in plants [43]. Increased expression levels of enolase and malate dehydrogenase in Δ*LeuRS* could promote the fungus to overcome salt. These genes were also up-regulated in halophytes exposed to hypersaline stress [44,45]. In addition, activation of glyoxylate and dicarboxylate metabolism was shown in sesames for tolerance to salinity [46]. Mutants under long-term salinity stress markedly increased the expression of genes coding catalase (which catalyzes the dismutation of H_2_O_2_ into H_2_O and O_2_) but almost abolished the expression of (S) -2-hydroxy acid oxidase (which transforms glycolate into H_2_O_2_ and glyoxylate), suggesting that the elimination of reactive oxygen species is important for fungal osmoadaptation. And the salt-induced WT samples up-regulated many compatible sugars (e.g., sucrose, D-mannose, and D-mannose 1-phosphate) (Figure 5), suggesting that soluble sugars could act as osmoprotectants in the fungus. 

We found that some genes encoding transcription factors (such as the Zn (II) 2 Cys6 (C6) transcription factor, BrlA, and HOG1), maintained relative low expression in Δ*LeuRS* (Appendix A). C6-like transcription factors regulated the metabolism of branched-chain amino acids (leucine, isoleucine, and valine) in *A*. *fumigatus* and sexual development in *N*. *crassa* [47,48]. BrlA, a central regulator, activates the conidial developmental pathway of *A*. *nidulans* [13]. And activation of the HOG1-mediated pathway promotes osmotic tolerance and influences sexual reproduction of fungi [49,50]. In addition, Δ*LeuRS* varied the expression levels of other reproduction-related genes in aspergilli. The core ‘velvet’ proteins (VeA, VelB, and LaeA) play a pivotal role in light and dark regulation of sexual development [51]. The removal of *VeA* increased the sensitivities to high osmolarity, oxidation stress, and triggered the production of fungal conidia [52]. Collectively, our results suggested that *LeuRS* modulated the metabolism of some amino acids and fungal reproduction by regulating the activities of relative transcription factors.

## 5. Conclusions

We constructed the Δ*LeuRS* mutant and characterized the cellular function of *LeuRS* in *A*. *montevidensis* under osmolarity conditions. Our results demonstrated that the deletion of *LeuRS* influenced the expression of multiple genes of the halophilic fungus. We observed that the mutants varied the expression levels of many genes that govern reproduction processes in the fungus (Figure 6a). We found that DEGs of Δ*LeuRS* mutants and WT strains were significantly enriched in KEGG pathways such as ABC transporters, alanine, aspartate and glutamate metabolism, arginine and proline metabolism, ornithine cycle, and sugar metabolism (Figure 6b). Metabolomic analyses showed that a range of metabolites varied significantly in this pairwise comparison, such as amino acids, sugars, and organic acids. The findings will help reveal the osmoadaptation strategies of *A*. *montevidensis* under high saline conditions.

## Figures and Tables

**Figure 1 jof-10-00036-f001:**
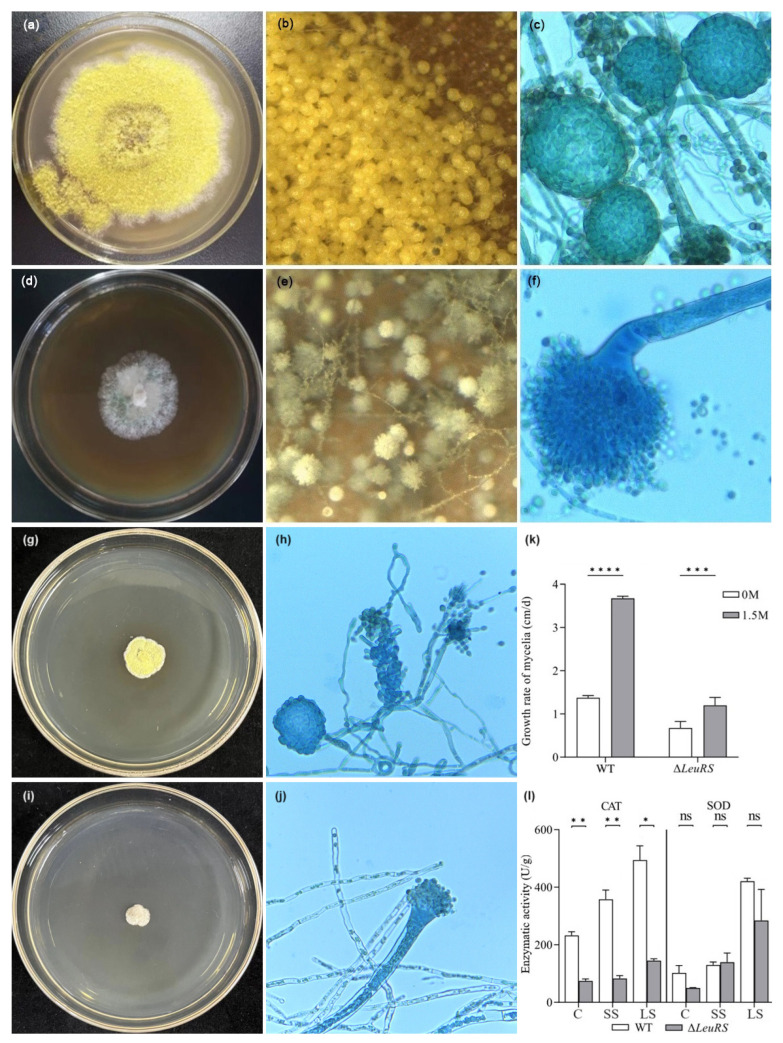
Morphological characteristics, growth rates, and antioxidative activities of the wide-type *A. montevidensis* (WT) and Δ*LeuRS* mutant grown on YPD with and without 1.5 M NaCl at 28 °C for 14 days. Under saline conditions, colony morphology of WT (**a**) and Δ*LeuRS* mutant (**d**), in situ observation of the microscopic morphological characteristics of WT (**b**) and mutant (**e**), cleistothecia of WT (**c**) and conidial heads of Δ*LeuRS* (**f**). Colonial characteristics and microscopic characteristics of WT (**g**,**h**) and Δ*LeuRS* (**i**,**j**) grown on YPD without NaCl. Growth rates (**k**) and antioxidant activities (**l**) of WT and Δ*LeuRS* mutants (*, *p* < 0.05, **, *p* < 0.01, *** *p* < 0.001, ****, *p* < 0.0001, ns, *p* > 0.05).

**Figure 2 jof-10-00036-f002:**
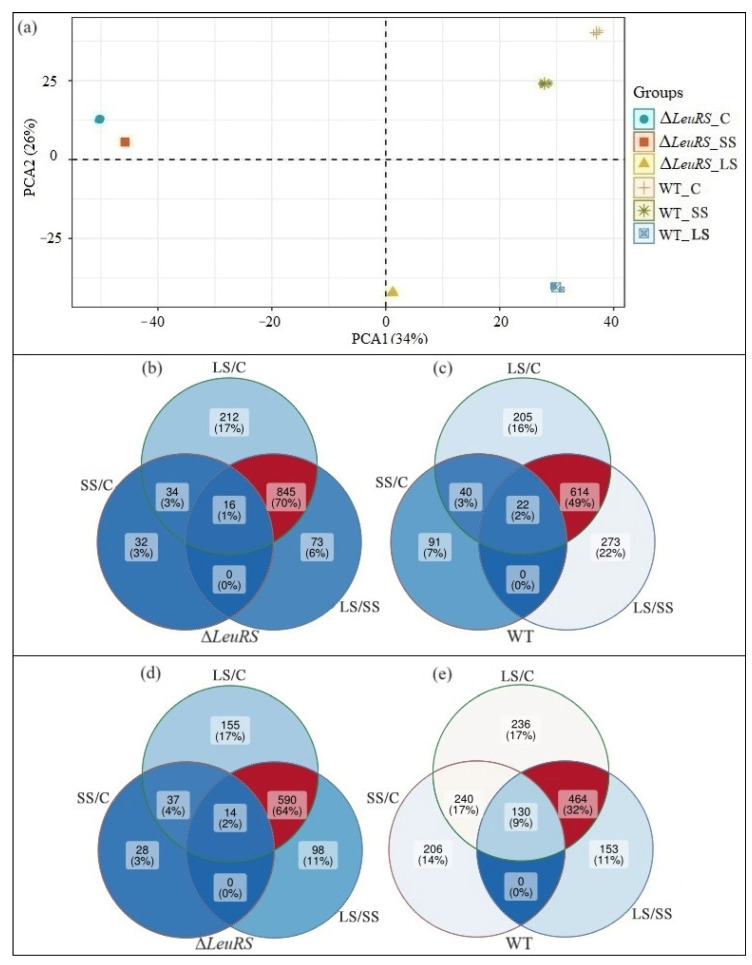
PCA (**a**) and Venn analysis of genes up- (**b**,**c**) and down-expressed (**d**,**e**) in Δ*LeuRS* mutants and WT strains under different saline conditions (C = control, SS = short-term salt stress, LS = long-term salt stress).

**Figure 3 jof-10-00036-f003:**
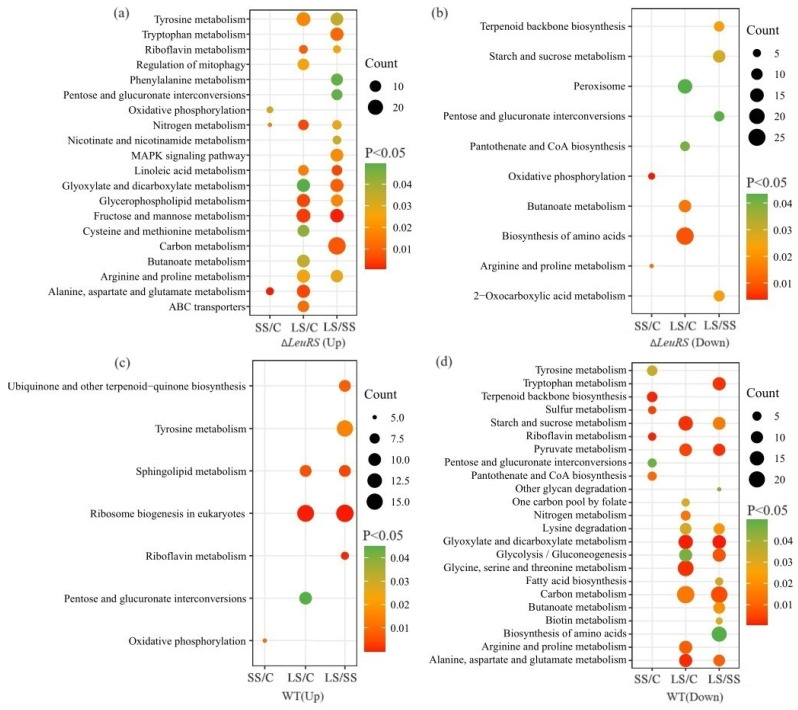
KEGG enrichment analysis of genes up- and down-expressed in Δ*LeuRS* (**a**,**b**) and WT (**c**,**d**), respectively (C = control, SS = short-term salt stress, LS = long-term salt stress).

**Figure 4 jof-10-00036-f004:**
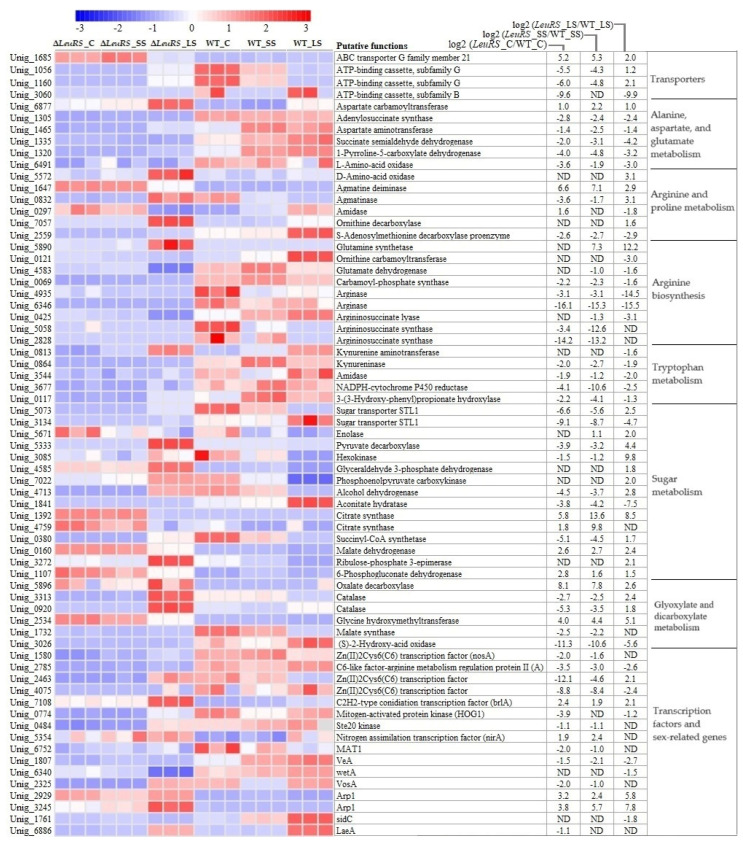
Heat map analysis of relative differential expression levels of some DEGs in compared groups of Δ*LeuRS* mutants and WT strains under distinct osmotic stress conditions.

**Figure 5 jof-10-00036-f005:**
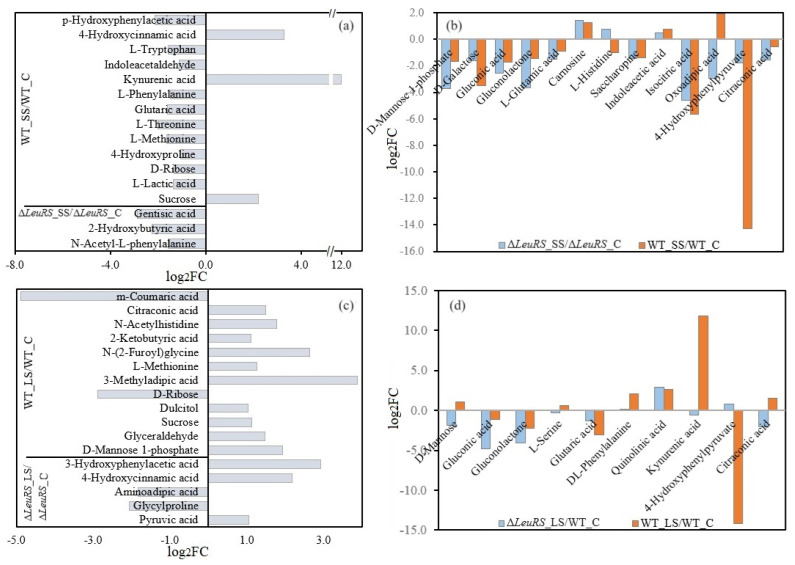
Analysis of discriminatory metabolites (*p* < 0.05, log2FC ≥ 1.0). The relative abundances of compounds identified only in Δ*LeuRS*_SS/Δ*LeuRS*_C and WT_SS/WT_C (**a**) and molecules co-occurred in these two groups (**b**). The relative contents of the unique compounds detected only in Δ*LeuRS*_LS/Δ*LeuRS*_C and WT_LS/WT_C (**c**) and compounds shared by these two groups (**d**).

**Figure 6 jof-10-00036-f006:**
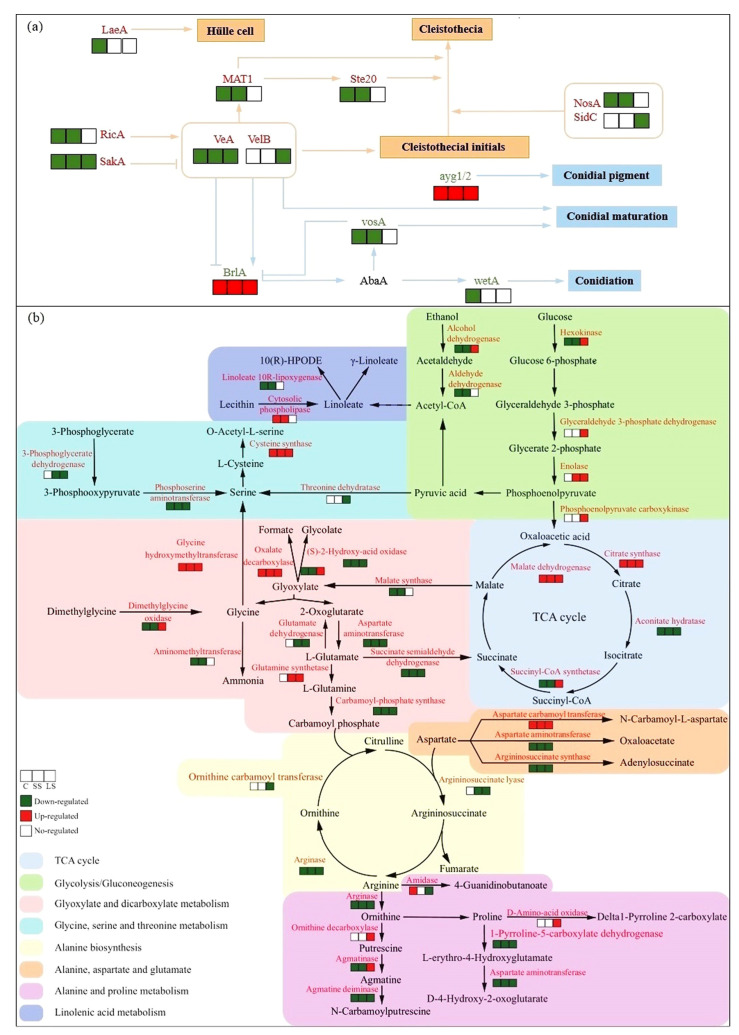
Conceptual profile showing enriched pathways of DEGs and metabolites between Δ*LeuRS* mutants and WT strains. DEGs involved in sexual and asexual processes (**a**). DEGs and metabolites were related to the metabolic pathways of amino acids, sugars, organic acids, and fatty acids (**b**). Genes marked with red or green color showed that these were up- and down-regulated at different time points.

## Data Availability

The data of all results in this study are provided in the manuscript and Appendix A.

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
