# Peer review of "The Deletion of LeuRS Revealed Its Important Roles in Osmotic Stress Tolerance, Amino Acid and Sugar Metabolism, and the Reproduction Process of Aspergillus montevidensis"

_jof, 2024, doi:10.3390/jof10010036_

Round 1

Reviewer 1 Report

Comments and Suggestions for Authors

The manuscript contains a lot of interesting and useful information that will be of interest to a wide range of readers. The experimental approaches used are appropriate. However, the presentation and interpretation of the results needs improvement.

General remarks

1. Since leucyl-tRNA synthase is an essential enzyme, authors should briefly summarize why deletion of the leuRS gene is not lethal in A. montevidensis.

2. LeuRS is required for translation and may also have a regulatory role. A strong decrease in LeuRS activity may cause changes in fungal physiology not only due to inappropriate regulation of certain processes, but also due to impaired synthesis of proteins (especially those rich in leucine).

3. Assuming that LeuRS plays a regulatory role, some of the differences recorded between the gene deletion mutant and the reference strain can be explained by the lack of this regulatory function in the mutant (direct effects) and some of the others are the result of cells trying to compensate for the negative consequences of the direct effects (indirect effects). It is not clear from the manuscript what the direct and indirect consequences of the gene deletion might be.

4. It is not clear from the manuscript whether the deletion of the leuRS gene had strong consequences at high osmolarity because LeuRS is involved in the regulation of processes that are particularly important for adaptation to high osmolarity, or whether the missing enzyme disrupted several biological processes that are usually important for cell function (regardless of osmolarity).

Specific remarks

Materials and methods

- lines 113-114 “Genes with a false discovery rate (FDR < 0.001) and log2 ratio ≥1 were identified as differentially expressed genes (DEGs) [23].”

This sentence is not clear. Suggestion: Genes with a false discovery rate (FDR) < 0.001 and |log2 ratio| ≥1 were identified as differentially expressed genes (DEGs) [23].

- line 142. “2.5. Statistical analysis“

This chapter is not consistent. The evaluation of the transcriptome data is presented in chapter 2.3, while the evaluation of the metabolome data is presented here.

- line 144  ”Student's t-test was performed to examine the significant differences (P<0.05)…”

It is unlikely that a Student's t-test (especially without p-value adjustment) was performed for all experiments. The authors should provide the experiments where a Student's t-test was used.

Results

- Fig. S2 and Fig. S4

The results of the statistical evaluation are not shown in these figures.

- According to Fig. S2, A. montevidensis is osmophilic (grows better at higher than normal osmolarity) and not simply osmotolerant.

line 334 “To reveal the functional roles of LeuRS (which was strikingly up-expressed in A. montevidensis under osmotic stress), we constructed a knockout mutant of LeuRS and found that the ΔLeuRS mutant of A. montevidensis retarded mycelial growth rate, decreased salt tolerance, and deactivated cleistothecium formation compared to the WT strain (Fig. 1, Fig. S2).”

A statistical analysis (e.g. two-way ANNOVA) is needed to confirm that the leuRS gene deletion retarded stronger the growth in the presence of NaCl than in the absence of it. Alternatively, the authors should simply conclude that deletion of the gene inhibited growth both in the presence and absence of NaCl.

- Fig. S3

According to Panels A and B the mutant had bigger colonies than the reference strain. In contrast, Fig. S2 suggests that the reference strain grew faster than the mutant. If the titles of Panel A and B are correct, this discrepancy should be explained.

- line 163 “Mycelial growth showed a significant morphological difference…”

The term "significant" should only be used if supported by statistical tests.

- line 174 “These results indicated that LeuRS plays a crucial role in the regulation of mycelial growth, antioxidative capability, and sexual development of A. montevidensis.”

This statement is misleading. These valuable data show only that the absence of LeuRS substantially affected the studied parameters. The effect can be either direct or indirect.

- line 184 “With three biological replicates, transcriptomic sequencing generated 383,214,494 and 399,020,304 clean reads for the three comparison groups (18 libraries).”

It is not clear what the two numbers represent and why they are important. The number of sequenced reads and the percentage of the reads assigned to the reference genome in the case of each transcriptome would be more informative. These data could be provided in a supplementary table; alternatively, they could be summarized briefly in the Materials and methods section.

- lines 191-197

No need to define DEGs again if they have been defined earlier. No need to repeat the numbers which are shown on the figure (Fig. 2B).

- line 197 “Our results showed that LeuRS was of great significance in mediating gene expression in A. montevidensis in response to the change in salinity.”

The presented data do not show that salinity was an important factor. There were large differences between the transcriptomes of the two strains even in the absence of NaCl, and the magnitude of the difference did not increase after treatments. Figure S3 also shows that deletion of the leuRS gene had strong consequences in the absence of NaCl. Fig 2B should be replaced with a Venn-analysis showing both the total number of enhanced and repressed genes in each comparison and the overlaps between them. The small overlaps would suggest the two strains responded differently to the treatments. It would also be useful to see whether or not the reference strain and the mutant had similar responses to high osmolarity. For example, the authors should show how much overlap there was between the DEGs of the WT_1h/WT_0h and deltaLeuRS_1h/ deltaLeuRS_0h comparisons. Alternatively, it could be shown how strongly the log2ratio values of the WT_1h/WT_0h and deltaLeuRS_1h/ deltaLeuRS_0h comparisons were correlated with each other.

- The authors have previously investigated the transcriptomic consequences of salt treatment (doi.org/10.1007/s00253-019-09705-2). It would be interesting to see whether or not the transcriptional changes recorded with the reference strain in the previous study and the present study are similar.

- Title of Figure 3

It is not clear what the “a-e” letters indicate.

- Figure 4

Authors should specify the type of data they are presenting (e.g. RPKM).

- Metabolomic data

Any information on the concentration of the detected metabolites, if available, would be useful. If the concentration of a metabolite (or the transcriptional activity of a gene) is very low in the reference strain, even a relatively high fold change value caused by deletion of the gene is of little biological relevance.

The presence of D-Ala-D-Ala dimers in an Aspergillus species is not self-evident, therefore it should be explained briefly.

Reviewer 2 Report

Comments and Suggestions for Authors

Peer review of Deletion of LeuRS influenced osmotic stress tolerance, nitrogen and carbon metabolism and reproductive process of Aspergillus montevidensis.

Major comments:

The title is too ambitious and does not reflect upon the observations show in the manuscript. For example, no data is shown that LeuRS is actually influencing nitrogen or carbon metabolism. Some nitrogen metabolic genes are downregulated but than can also be caused by nitrogen starvation. The same for carbon metabolism.

The leucyl-tRNA synthetase (leuRS) deletion mutant is showing phenotypes when no salt-stress is present making it difficult to claim that salt stress is causing a phenotype when leuRS is deleted. leucyl-tRNA synthetase are known to be involved in primary functions as protein synthesis, deletion of a crucial gene can result in poor protein synthesis resulting in affected cellular functions, structures and activities. LeuRS was shown to be induced by salt-stress but this can also induced because of the need of proteins related to salt stress.

A strain that already has developmental issues will likely have more stress from any challenging growth condition compared to a healthy strain. This is also supported by the transcriptome data (fig 2) which shows that the WT and LeuRS mutant are different from the start at timepoint 0h without salt stress. Supplemental figure S2 and S3 is essential data since it shows that the deletion of leuRS is also causing phenotypes without salt stress, it should be included as figure, preferably combined with figure 1, in the manuscript.

This also causes my major concerns about the correctness of the experimental setup used for the transcriptomics for multiple reasons.

1.       The strains are grown for 14 days with NaCl (long-term salt stress), 14 days without NaCl (0h control) and 14 days without NaCl and then 1h with 1.5M NaCl (short-term salt stress). What was the pH of the cultures? How was the state of the mycelia? How much biomass was formed (dry weight)? These are important factors for the condition of the mycelia.

2.       The strains were grown for 14 days in YPD liquid media, as carbon source 2% glucose was added (+ 2% dextrose from YPD). Due to the long growth time, the strains were probably in a carbon starvation/limited state which is also a stress condition and causes autophagy/cell lysis. The same counts for nitrogen starvations/limitation.

3.       The transcriptome data is compared using WT and deletion strain under the same condition. However, this is not a correct way since the morphology of the strains is completely different and the mutant is already growing differently compared to the parental strain without any stress condition. Therefore, there are already huge difference between the expressed genes. I would suggest to taken another approach to remove background noise by comparing the mutant 0h control with the mutant short-term stress and long-term stress transcriptome data and do the same for the parental strain (wildtype). Then, compare the difference in up/down regulated genes between the mutant and wildtype. Are there similar or different genes upregulated to the response of salt stress?

4.       Are there any fungal genes (homologs) known that are related to salt stress/tolerance response and were they also effected? Is leuRS expressed in the wildtype under your used conditions? Is there a difference in long-term and short-term NaCl exposure?

I am also curious why there were no chaperones or heat shock proteins upregulated during stress response? They are known to act upon osmotic stress.

I advise to compare the 0h control with the long-term stress and short terms stress metabolomes of the mutant and wildtype. This will show if the response of the WT and mutant are comparable. At this moment, the mutant is crippled, has less biomass and is probably in a different growth state than the wildtype.

Supplemental figure S2; I think there is a mix up in the supplemental fig 3, it shows that (3b) the wild type is growing poorer then the mutant (3a) which is contradicting. Based on the yellow cleistothecia, fig. S3a is the wildtype and fig. S3b is the mutant.

Line 114: In the materials and methods, a FDR of <0.001 is used. However later on, a FDR of <0.05 is used (line 192). Which one was used?

Line 121: How much mycelia was used? Was this even quantified? Without this quantitative comparisons cannot be made.

Table S1: Missing the RPKM and p-values in the tables. Did you also use a threshold for the RPKMs? This is advised in order to remove insignificant gene expressions.

Line 350-352: Your data shows that at timepoint 0h there are ABC transporters differently expressed (table S1) but when salt is added, these transporters are not differently expressed anymore. This would mean that either the response is still there and LeuRS is not vital for this process.

Line 378-381: Can you explain why you see proline and arginine genes are upregulated (which are involved in salt stress) but no arginine or proline is observed in the metabolome.

Line 408: L-tyrosine was observed in the metabolomics after 1h of salt induction (8.5x higher than the mutant) but there is no difference in L-tyrosine observed in long term salt stress.

Minor comments:

Line 16: p-values of transcriptome data are not show in the manuscript.

Line 27: The filamentous fungi Aspergillus….

Line 56-57: Aspergillus niger is an asexual fungus and is not able to produce cleistothecia, which is a sexual fruiting body. It is able to form sclerotium.

Line 56-57: Please cite the original papers, not a review.

Figure 1.: I would suggest to put the pictures from the wild-type above and the LeuRS mutant below. This way the comparison is more clear and less confusing.

Line 178: … for 14 days at 28 °C.

Figure 2.: DEGs numbers in the bars are not visible. Change the color or put them next to the bars. The same for figure 5.

Supplemental fig. 2: for how many days was this measured? Please add the growth time in the legend. The same for fig S3.

Figure 6.: Text in the figure is blurry. Please, increase the resolution of the figure.

Line 441-443: were any homologs of these genes differential expressed in the mutant or similar to the wildtype?

How where the plates inoculated with spores or agarplugs?

Round 2

Reviewer 1 Report

Comments and Suggestions for Authors

The manuscript improved a lot.

1. Authors should add their hypotheses to the manuscript why deletion of the leuRS1 gene was not lethal in A. montevidensis. If it is based on the leuRS2 gene, they should add a supplementary figure to the manuscript showing the mean ± SD RPKM values of both leuRS1 and leuRS2 genes for all the six groups (two strains, three time points) (Standing alone, log2FoldChange values are not informative in this case.)

2. As LeuRS is required for translation (in addition to its potential regulatory role), a strong decrease in LeuRS activity may cause changes in transcriptome/fungal physiology due to impaired translation of (e.g. regulatory) proteins. In the manuscript, the authors should summarize evidences supporting the regulatory role for LeuRS1 and exclude the possibility that the observed changes are a consequence of the translation of regulatory proteins becoming inadequate in the absence of LeuRS1. Alternatively, they should be much more cautious about the (direct) regulatory role of LeuRS1. E.g. they could hypothesize that some of the changes are a consequence of the regulatory role of LeuRS. But explaining why they think this is important even in this case.

lines 269-270: “Our results indicated that LeuRS are important for the metabolization of amino acids and sugars of halophilic fungi, especially in the long-lasting salinity condition.”

It can be accepted. However, the question is whether this is a consequence of inappropriate translation (causing reduced growth, reduced protein synthesis, reduced amino acid demand, etc.) or a consequence of a missing regulatory role of LeuRS1. Transcriptomic data may not be sufficient to answer this question.

lines 401-403: “Based on these findings, we conclude that LeuRS could play a vital role in the regulation of ABC transporter-mediated substance transport in the osmoadaptation of the halophilic fungus.”

lines 413-414: “These showed that LeuRS is important for the regulation of amino acid metabolization of A. montevidensis in response to salinity stress.”

lines 438-439: “These indicated that LeuRS influences the proline and arginine metabolic pathway to enhance fungal adaptation to salinity.”

lines 493-495: “According to our findings, it can be inferred that LeuRS is of importance for the regulation of sugar metabolism in the defense of fungi against salt stress.”

lines 505-506: “Taken together, our finding showed that LeuRS could act as a prominent modulator for reproduction process in aspergilli.

Transcriptomic changes only show that, for example, the transcription of ABC transporter genes, amino acid metabolism genes, carbohydrate metabolism genes was altered in the absence of LeuRS1. As mentioned above, this could be a consequence of the lack of regulatory function of LeuRS (as suggested by the authors), or a consequence of physiological changes caused by inadequate translation, or it could be a response to compensate for the deleterious consequences of the lack of LeuRS1 (indirect effect). Again, transcriptomic data may not be sufficient to answer this question. Improper translation of proteins can alter anything in the cells.

Authors wrote: We … characterized the cellular function of LeuRS in A. montevidensis under osmolarity conditions (line 509). In fact, the authors described the consequences of the missing LeuRS protein. Unfortunately, this is not equivalent to characterizing its function.

3. Regarding Figure 5, if the concentration of a metabolite is very low in the reference strain, even a relatively high fold change value is of little biological relevance. For example, if the concentration of D-ribose is 1 pmol/L in the reference strain and 1000 pmol/L in the mutant, the fold change value is 1000, but the biological consequences of 1000 pmol/L D-ribose are still questionable. The same fold change value with 0.1 mmol/L and 100 mmol/L concentrations, however, promises strong biological consequences. Therefore, Figure 5 is of little relevance unless the authors provide information on metabolite concentrations or at least the signal intensity values used to calculate the log2FoldChange values (e.g. as a supplementary file).

lines 427-430 “Furthermore, we observed that ΔLeuRS mutants compared to WT strains under saline conditions increased the levels of some amino acids and their derivatives (e.g., L-asparagine, L-glutamine, D-alanine, γ-glutamylalanine, N-acetylglutamine and D-alanyl-D-alanine) (Table S2), which could play a crucial role in improving salt tolerance of fungi”

lines 463-466 “Furthermore, deletion of LeuRS led to the differential accumulation of many metabolites (e.g., 4-hydroxyphenylpyruvate, kynurenic acid, L-phenylalanine, L-tyrosine, and quinolinic acid) from the tryptophan pathway in A. montevidensis under different salinity conditions (Fig. 5, Table S2). 4-hydroxyphenylpyruvate was dramatically accumulated (by a factor of more than 16 times) in salt-induced ΔLeuRS strains, compared to controls, suggesting a potential protective role of this compound for mutants against salt stress.”

These can be acceptable if the experimental data demonstrate that the concentration of these compounds is not infinitively small.

The “accumulation” of a compound does not necessarily mean that the cells produce that compound as an end product. It may be an intermediate of a pathway, and the increased level may indicate, for example, increased flux in the pathway. Therefore, it is also possible that it was not the “accumulated” metabolite but its derivative (which may not have been detected) that played a protective role. Please take it into consideration during the evaluation of the data.

Minor complains

1. lines 194-209: Instead of repeating the numbers in Figure 2, the authors should just briefly summarize the meaning of the figure.

2. lines 211-218: This is a long and confusing title.

3. lines 220-270: Instead of repeating the terms in Figure 3, the authors should just briefly summarize the meaning of the figure.

4. lines 272-276: This is a long and confusing title.

5. Figure 5 mentions several compounds. Among these, the D-Ala-D-Ala dimer (and perhaps others) is not a typical component of fungal cells. (For some readers, the presence of the D-Ala-D-Ala dimer in a fungal culture indicates that the culture was contaminated with bacteria.) Therefore, the authors should either add a reference to the occurrence of this metabolite in fungi, or at least a comment that the presence of this metabolite in this species is interesting/unexpected/requires further investigation.

6. Fig. 2 and lines 208-209: “Our results showed that LeuRS was of great significance in mediating gene expression in A. montevidensis in response to the change in salinity.”

(Just a suggestion and not binding advice:)

Omit the 14d/1h comparison from Fig. 2b,c,e,f and show only the transcriptional consequences of short and long term exposure. These figure will demonstrate that more genes responded to long term treatment than to short term treatment, but the majority of the genes responded to short term treatment also responded to long term treatment (as expected).

On Fig 2 d and g, the 0h/0h comparisons show the original difference between the two strains recorded before the treatments. If the two strains respond differently to the treatments this difference will change. The big overlap between the sets indicates that the responses of the two strains were more or less similar. It is true for the 0h/0h and 1h/1h sets suggesting that the short term treatment elicited similar changes in the two strains. However, in the case of the 0h/0h and 14d/14d sets the overlap is smaller suggesting that the responses for long term treatments were different.

Originally I suggested a Venn diagram that shows for example the overlap between the 1h/0h mutant (2b) and 1h/0h wt (2c) gene sets. This diagram would show that what percentage of the genes upregulated in the reference strain also upregulated in the mutant by the treatment. A small overlap may indicate that the two strains responded differently.

(Of course, the 1h/0h mutant (2e) and 1h/0h wt (2f), 14d/0h mutant (2b) and 14d/0h wt (2c), 14d/0h mutant (2e) and 14d/0h wt (2f) comparisons are also relevant.)

7. Fig. 5 (Just a suggestion and not binding advice:)

Most readers are interested in how fungi adapt to high osmolarity. Therefore showing the changes in metabolite concentrations after the treatments (1h/0h and 14d/0h) would improve the manuscript.

Author Response

Thank you for your comments. We done accordingly. Please find the attached file.

Reviewer 2 Report

Comments and Suggestions for Authors

Dear authors,

Thank you for the revised version, see the additional reviewer comments in the summited file

Author Response

(The authors gave the same response as above.)

Round 3

Reviewer 1 Report

Comments and Suggestions for Authors

The manuscript is acceptable in its current form.

Author Response

Thank you very much for your comments again.

Reviewer 2 Report

Comments and Suggestions for Authors

The manuscript improved drastically especially in the result and discussion section. However, I still have some minor improvements/comments.

Minor comments:

Line 246 and 411: Zn(II)2Cys6 (C6)

Line 280: remove: And

Line 282: Mannose 1-phosphate (Log2FC = -3.7)

Line 328: ‘’a mitochondrial leucyl-tRNA synthetase’’ is in a different font or size

Legend of figure 2 and 3 needs the abbreviations explained. (C = control, SS = short-term salt stress. LS = long-term salt stress)

Figure 4. Log2 labels should be on top of the figure.

Figure 5, axis labels are growing through the bars and y-axis. It would be better if the labels are underneath the bars

Author Response

Replies to the comments of Reviewer #2

Thank you very much for your positive comments. According to your suggestions, we revised the manuscript again. Please find the detailed responses below and the corresponding revisions in track changes (marked in blue) in the re-submitted manuscript.

Minor comments:

Line 246 and 411: Zn(II)2Cys6 (C6)

Response: Done accordingly.

Line 280: remove: And

Response: Deleted.

Line 282: Mannose 1-phosphate (Log2FC = -3.7)

Response: Changed.

Line 328: ‘’a mitochondrial leucyl-tRNA synthetase’’ is in a different font or size

Response: Changed the front.

Legend of figure 2 and 3 needs the abbreviations explained. (C = control, SS = short-term salt stress. LS = long-term salt stress)

Response: Done accordingly.

Figure 4. Log2 labels should be on top of the figure.

Response: Done accordingly.

Figure 5, axis labels are growing through the bars and y-axis. It would be better if the labels are underneath the bars

Response: Thank you for your suggestion. We weren’t able to change it accordingly although we tried many times.